# Maximum Mean Discrepancy for Generalization in the Presence of Distribution and Missingness Shift

## Abstract

Covariate shifts are a common problem in predictive modeling on real-world problems. This paper proposes addressing the covariate shift problem by minimizing Maximum Mean Discrepancy (MMD) statistics between the training and test sets in either feature input space, feature representation space, or both. We designed three techniques that we call MMD Representation, MMD Mask, and MMD Hybrid to deal with the scenarios where only a distribution shift exists, only a missingness shift exists, or both types of shift exist, respectively. We find that integrating an MMD loss component helps models use the best features for generalization and avoid dangerous extrapolation as much as possible for each test sample. Models treated with this MMD approach show better performance, calibration, and extrapolation on the test set.

## 1 Introduction

Machine learning models often work on the assumption that the training and test datasets follow the same distribution. In practice, this might not be the case, yet the difference in distribution, called dataset shift, is often ignored. This might be acceptable when the dataset shift is small, but small scale shift is not always guaranteed. When it is not, it can lead to poor performance on shifted test sets. Dataset shift can be divided into three main categories: covariate shift, prior probability shift, and concept shift. Covariate shift occurs when the distribution of input features changes, prior probability shift occurs when the distribution of target variables changes, and concept shift occurs when the relationship between input features and target variables changes (Quinonero-Candela et al., 2008). In this paper, we focus on covariate shifts.

There have been some attempts to address covariate shift in previous literature. One straightforward approach is to identify the shifted covariates and drop them from the feature set. However, this can be challenging for high-dimensional data, and one may also lose useful information in the process. A softer method is to use importance re-weighting by assigning higher weights to training samples that are more similar to the samples in the test dataset. The importance weights can be calculated through various techniques, including density estimation (Shimodaira, 2000; Kanamori & Shimodaira, 2000; Zadrozny, 2004; Dudík et al., 2005), kernel mean matching (Huang et al., 2007), Kullback-Leibler importance estimation (Sugiyama et al., 2008), and discriminative learning (Bickel et al., 2009). These methods work particularly well for sample selection bias, but they treat samples at a row/sample level which is not flexible enough.

In this paper, we propose optimizing an auxiliary maximum mean discrepancy (MMD) loss with a mixture of kernels to treat the covariate shift problem. MMD is already widely used in unsupervised domain adaptation (UDA) (Tzeng et al., 2014; Long et al., 2015; 2017; Yan et al., 2017; Chen et al., 2020). Here we extend it to the more general covariate shift problem. Conceptually, we can think of domain adaptation as an extreme case of covariate shift as the joint distribution of input variables shifts to a different domain entirely.

Moreover, previous UDA works has never considered an important special case of covariate shift that involves data missingness. For example, in an internal project of carbon emission prediction in Section 4.3, we try to train a model with the companies who reported carbon emission to infer carbon emission for the unreported companies. However, we find that the unreported companies usually

have higher missing rates in various sets of features compared to the reported companies. A similar situation can happen in other use cases. For instance, various physical measurements and clinical exams need to be taken before a diagnostic decision of a disease can be made, but when estimating the probability of risk of the disease on a new group of people, different measurements and exams can be missing for different subgroups. Or, in financial fraud detection, the unlabeled set may also have higher feature missing rates compared to the labeled set, especially in the self-reported features. If we simply ignore the data missingness shift, the model performance may be degraded on the unlabeled set. In this paper, we propose building a novel masker model to mask training samples using MMD and thereby align the missingness patterns in the training and test datasets. We find that this can help improve the test performance by preventing the model from depending on relationships that will be unavailable at test time. Also, note the MMD masker model is especially relevant when categorical features are involved, as new categories in the test set correspond to missing features in the training set, see Section 4.2 for an example.

Expanding on this, we combine a missingness shift treatment and a general feature distribution shift treatment together in a hybrid model, and show that the hybrid model is superior through various experiments done with both synthetic data and real data.

## 2 RELATED WORK

MMD was first introduced in the two-sample tests (Gretton et al., 2007). It is a non-parametric metric to measure the distance between two distributions. Therefore, to compare two distributions, one does not need to know their probability density functions, or PDFs. This is useful as the PDFs can be difficult to calculate or can even be unknown. Because of its simplicity, MMD has been widely applied to various problems, such as goodness-of-fit testing (Jitkrittum et al., 2018), MMD GAN (Li et al., 2015; Sutherland et al., 2017; Li et al., 2018), and MMD VAE (Zhao et al., 2019).

One technique related to using MMD for covariate shift is using kernel mean matching to re-weight training samples (Huang et al., 2007). In that paper, the authors formulated a quadratic problem between empirical means to find suitable weights for training samples, after which they could apply weighted ordinary least squares or support vector machines. The difference is that they tackled the problem from the importance resampling angle and used the results for more traditional machine learning methods. We incorporate MMD directly into a neural network model in one step instead of two by using the representation learning ability of neural networks, and are able to treat covariate shift in a more granular level than the row/sample level if necessary.

The most related application of MMD to our work is in UDA. Deep Domain Confusion (DDC) (Tzeng et al., 2014) is one of the earliest attempts to use MMD for domain invariant learning. Then, Deep Adaptation Networks (DAN) (Long et al., 2015) were proposed to match an optimal multi-kernel MMD to reduce domain discrepancy. This was followed by Joint Adaptation Networks (JAN) (Long et al., 2017), which were proposed to apply joint MMD in multiple domain-specific layers across domains. Yan et al (2017) proposed adjusting plain MMD by class-specific auxiliary weights to account for the class weight bias across domains. Chen et al (2020) proposed matching higher-order moments to perform fine-grained domain alignment. Note that these applications all focused on domain shifts in classification problems, while we extend the application to more general distribution shift and to missingness shift that has never been considered before.

## 3 METHOD

### 3.1 PRELIMINARY: MAXIMUM MEAN DISCREPANCY

Assume we have two sets of samples $X = \{x_i\}_{i=1}^{N}$ and $Y = \{y_j\}_{j=1}^{M}$ drawn from two distributions $P(X)$ and $P(Y)$. MMD is a non-parametric measure to compute the distance between the two sets of samples in mean embedding space. Let $k$ be a measurable and bounded kernel of a reproducing kernel Hilbert space (RKHS) $\mathcal{H}_k$ of functions, then the empirical estimation of MMD between the two distributions in $\mathcal{H}_k$ can be written as

$$\mathcal{L}_{MMD^2}(X,Y) = \frac{1}{N^2}\sum_{i=1}^{N}\sum_{i'=1}^{N}k(x_i,x_{i'}) + \frac{1}{M^2}\sum_{j=1}^{M}\sum_{j'=1}^{M}k(y_j,y_{j'}) - \frac{1}{NM}\sum_{i=1}^{N}\sum_{j=1}^{M}k(x_i,y_j)$$

(3.1)

When the underlying kernel is characteristic (Fukumizu et al., 2008; Sriperumbudur et al., 2010b), MMD is zero if and only if $P(X) = P(Y)$ (Gretton et al., 2012a). For example, the popular Gaussian RBF kernel, $k(x, x') = \exp(-\frac{1}{2\sigma^2}|x - x'|^2)$, is a characteristic kernel and was widely used in previous literature (Li et al., 2015; Sutherland et al., 2017; Long et al., 2017; Li et al., 2018). Dziugaite et al (2015) compared the RBF kernel with the rational quadratic (RQ) kernel and the Laplacian kernel and found that the RBF kernel performed best. Therefore, in this work, we also used the RBF kernel.

What remains is how to select the bandwidth parameter $\sigma$ in the Gaussian RBF kernel, which can affect the hypothesis testing power of MMD measurement. There were some heuristics in previous literature (Sriperumbudur et al., 2010a; Gretton et al., 2012b), but later research (Li et al., 2015; Sutherland et al., 2017; Li et al., 2018) found that using a mixture of 5 or more bandwidths performed well, so we followed this procedure as well.

## 3.2 MMD REPRESENTATION

Let us first define the context of our predictive problem. Assume that we have training and test features $X_{tr}, X_{te} \in \mathcal{R}^m$, where $m$ is the number of features. We also know the target value $y_{tr}$ for the training set, and we want to build a model to predict $y_{te}$ for the test set. When we train the model with $(X_{tr}, y_{tr})$, a covariate shift between $X_{tr}$ and $X_{te}$ can cause the trained model to perform poorly on the test set.

Assume we build a neural network model for the task, then similar to what has been widely used in domain adaptation, we can match the MMD statistics between $emb_{tr}$ and $emb_{te}$ at some intermediate embedding layer so that the learned embeddings have similar distributions between the training and test sets and will therefore reduce the amount of extrapolation. We call this approach MMD Representation. See Figure 1 for the architectural details. Note that models like DAN (Long et al., 2015) and JAN (Long et al., 2017) can be considered as special forms of MMD Representation, as they also apply MMD to multiple specific intermediate layers. The final loss we will optimize is the original task loss plus the MMD loss in the representation space:

$$\mathcal{L} = \mathcal{L}_{task} + \lambda \mathcal{L}_{MMD^2}(emb_{tr}, emb_{te}) \tag{3.2}$$

where $\lambda$ is a hyperparameter to adjust the ratio between the original loss and the MMD loss. Generally, we found that setting $\lambda$ to match the magnitudes of the two loss terms leads to the best results. Compared to the approach of dropping features with covariate shift before training, MMD Representation is able to retain as much information as possible from each individual feature and find a common feature space that has as little shift as possible between the training and test sets.

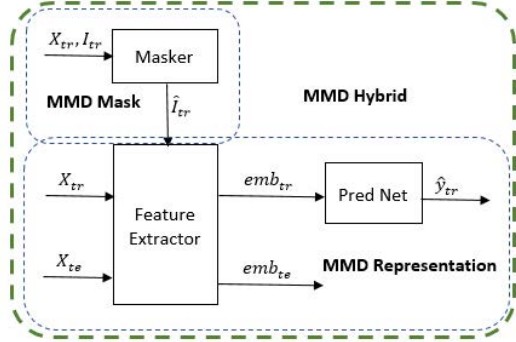

Figure 1: Architecture for MMD Representation, MMD Mask and MMD Hybrid. The two blue dashed boxes contain the architectures for MMD Mask and MMD Representation respectively, and the green dashed box contains the architecture for MMD Hybrid, which combines MMD Mask and MMD Representation.

---

**Algorithm 1** MMD Hybrid

**Input:** $X_{tr}, I_{tr}, X_{te}, I_{te}, y_{tr}, \lambda$

Initialize $f_M, f_F, f_P$

**for** each epoch **do**
    **for** each batch **do**
        $\hat{I}_{tr} \leftarrow f_M(X_{tr}, I_{tr})$
        $X'_{tr} \leftarrow$ Mask $X_{tr}$ by $\hat{I}_{tr}$
        Update $f_M \leftarrow$
                $\min \mathcal{L}_{MMD^2}((X_{tr}, \hat{I}_{tr}), (X_{te}, I_{te}))$

        $emb_{tr}, emb_{te} \leftarrow f_F(X'_{tr}), f_F(X_{te})$
        $\hat{y}_{tr} \leftarrow f_P(emb_{tr})$
        Update $f_F, f_P \leftarrow$
                $\min \mathcal{L}_{task}(\hat{y}_{tr}) + \lambda \mathcal{L}_{MMD^2}(emb_{tr}, emb_{te})$
    **end for**
**end for**

### 3.3 MMD MASK

Despite the advantages of MMD Representation, we found that it is insufficient when a missingness shift is involved, as was the case for the carbon estimation problem mentioned in Section 1. In general, our MMD Mask can be effective when the missingness rates for some set of features are low or zero in the training set but much higher in the test set. The architecture of MMD Mask is shown in Figure 1. Denote $I_{tr}$ and $I_{te}$ as the original missingness indicators for the training and test features $X_{tr}$ and $X_{te}$, respectively; we want to learn a new missingness indicator $\hat{I}_{tr}$ conditioned on $X_{tr}$ and $I_{tr}$ that can minimize the MMD loss between the joint distributions $P(X'_{tr}, \hat{I}_{tr})$ and $P(X_{te}, I_{te})$:

$$\mathcal{L} = \mathcal{L}_{MMD^2}((X'_{tr}, \hat{I}_{tr}), (X_{te}, I_{te})) \tag{3.3}$$

where $X'_{tr}$ is generated by masking the original $X_{tr}$ using learned $\hat{I}_{tr}$. The downstream model is then trained using $X'_{tr}$ instead of $X_{tr}$. In this way, the downstream model can focus on the right features for each training sample. Importance resampling using kernel mean matching also operates in the original feature space, but it removes or downweights each sample as a whole (at the row level), while we treat each sample feature by feature. Compared to MMD Representation, another advantage of MMD Mask is that it can still be applied when the downstream model cannot learn an intermediate representation (e.g. a tree-based model).

### 3.4 MMD HYBRID

Furthermore, we can combine MMD Representation and MMD Mask together as a more comprehensive shift treatment. We call this MMD Hybrid: see Figure 1 for its architecture. Denote the masker model as $f_M$, the feature extractor as $f_F$, and the prediction network as $f_P$. We use an alternating update method to train the MMD Hybrid model. That is, first update the masker model $f_M$ by minimizing the loss in Eq. (3.3) and then update the feature extractor $f_F$ and prediction network $f_P$ together by minimizing the loss in Eq. (3.2); see Algorithm 1 for details.

## 4 EXPERIMENTS

The experiments are designed to answer the following questions: (1) How do MMD Representation, MMD Mask and MMD Hybrid work with the presence of distribution shift and missingness shift in features respectively? Which MMD model performs best in which scenario? (2) Do MMD Representation, MMD Mask and MMD Hybrid work in real world datasets generally? How do they perform compared to other existing methods? (3) When the downstream model cannot learn an intermediate representation, do any of the MMD models still work?

To answer the questions above, we tested our approaches on both synthetic data and real data. The real data is comprised of a Bike Sharing dataset from the UCI Machine Learning Repository[1], an IEEE-CIS Fraud Detection dataset from Kaggle[2], an internal project of carbon estimation, and the image dataset MNIST (LeCun et al., 1998).

For all experiments except carbon estimation and MNIST, the baseline model is a multilayer perceptron (MLP) model. MMD Representation always shares the same architecture, but also matches MMD statistics at the last hidden layer. MMD Mask outputs masks for each feature per sample; we then mask out training samples based on the learned masks, and use that to train the same downstream baseline model. MMD Hybrid combines MMD Representation and MMD Mask. For synthetic data, the Bike Sharing data and the IEEE-CIS Fraud Detection data, the baseline models take features and missing indicators as inputs, and the missing values were imputed using the means from the training set. We didn't impute missing values for carbon estimation because it used a tree-based model that can handle missing values inherently. For all datasets, we use MMD statistics under a mixture of Gaussian RBF kernels with bandwidths of $\{1, 2, 4, 8, 16, 32\}$. The MMD Mask model uses a Relaxed Bernoulli distribution parametrized by a temperature $\tau$ to generate a mask probability for each feature. The Relaxed Bernoulli is a continuous approximation to a Bernoulli distribution over $(0, 1)$, similar to the Gumbel-Softmax trick (Jang et al., 2017) for categorical distributions. In all experiments, we set $\tau$ to 0.1 at the start of training and then gradually decrease it to 0.01. We

---

[1] https://archive.ics.uci.edu/ml/datasets/bike+sharing+dataset.
[2] https://www.kaggle.com/c/ieee-fraud-detection/overview.

implemented all models in PyTorch. More details about model architectures and hyperparameters can be found in Appendix A.

For all experiments, whenever possible, we also compared our methods with kernel mean matching (KMM) (Huang et al., 2007), and UDA baselines like DAN (Long et al., 2015) and JAN (Long et al., 2017). As for KMM, we followed the procedure in their Empirical KMM optimization Section, and found the best weights for training samples by solving the quadratic problem using cvxopt package[3], then used these weights to reweight samples in the original loss function. Whenever DAN and JAN are used, we apply MMD or Joint MMD to the last two hidden layers as MLPs in this paper are only three to four hidden layers.

## 4.1 SYNTHETIC DATA

In this experiment, we used synthetic data to gain a general understanding of which MMD models work best in which situations. We constructed a synthetic dataset as shown in Figure 2. There are two input features $X_1$ and $X_2$, both having some predictive power for the target $y$. The relationship between feature $X_1$ and $y$ is linear while the relationship between $X_2$ and $y$ is parabolic. We injected noise $\epsilon_1 \sim \mathcal{N}(0, 0.1)$ into feature $X_1$ and larger-scale noise $\epsilon_2 \sim \mathcal{N}(0, 0.5)$ into feature $X_2$. Based on this alone, the model should prefer using feature $X_1$ to feature $X_2$. However, we also engineered two types of data shifts into feature $X_1$ between the training and test datasets. We first resampled from the training dataset with replacement to get our test dataset such that both the training and

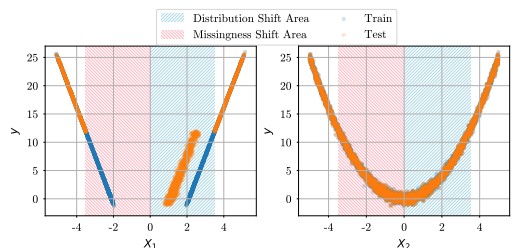

Figure 2: Synthetic data: plots of target y v.s. feature $X_1$ and $X_2$ respectively. Distribution shift of $X_1$ occurs in blue shaded area and missingness shift of $X_1$ occurs in pink shaded area.

test datasets had 5,000 rows. Then in the test dataset, we designed a missingness shift for $X_1$ in the range $[-3.5, 0]$ by masking it out, and a distribution shift for $X_1$ in the range $(0, 3.5]$ by subtracting it by $\epsilon \sim \mathcal{N}(1, 0.1)$ - see Figure 2. Therefore, the test dataset has both missingness and distribution shifts compared to the training dataset. We hope that a properly designed model can learn to predict using feature $X_2$ instead of $X_1$ in the regions of $X_1$ affected by the shifts.

Table 1 shows the average and standard deviation of mean squared error (MSE) for each model from 10 runs on the test set. As we can see, the performances of baseline model and KMM are similar. This is expected because KMM is an importance re-weighting technique that downweights training samples different from test samples at row level while what we hope to see is that a method can learn to ignore $X_1$ but use $X_2$ for those shifted training samples. On the other hand, all MMD methods improve the performance on the shifted test set to some degree. Among the methods to match the intermediate representations, DAN and JAN perform better than MMD Representation as they try to match more intermediate layers. But their performance are not as good as the MMD Hybrid model. Note that the performance of MMD Hybrid is very close to the golden model that is the true mean function used to generate this synthetic data.

Figure 3 gives more details of how each model performs in different ranges of $X_1$ and $X_2$ on the test data. We focus on the comparison among the basic baseline model, the MMD Representation model, the MMD Mask model and the MMD Hybrid Model here. The performance details for the KMM, DAN and JAN can be found in Appendix B.

In the region where data shift does not exist (areas without shades), all models learn the relationship very well and have very small residuals. In the region where missingness shift exists (areas with prink shades), since $X_1$ is masked out in the test dataset, we focus on residuals vs. $X_2$. Although the MMD Representation model already reduced residuals significantly compared to the baseline model, it still has higher residuals in this region compared to the MMD Mask model and the MMD Hybrid model. This indicates that when there is only missingness shift, MMD Mask is sufficient as it is a natural solution to missingness shift. In the region where distribution shift exsits (areas with blue shades), we see that the baseline model is worst, followed by MMD Mask, MMD Representation and

---

[3]https://cvxopt.org.

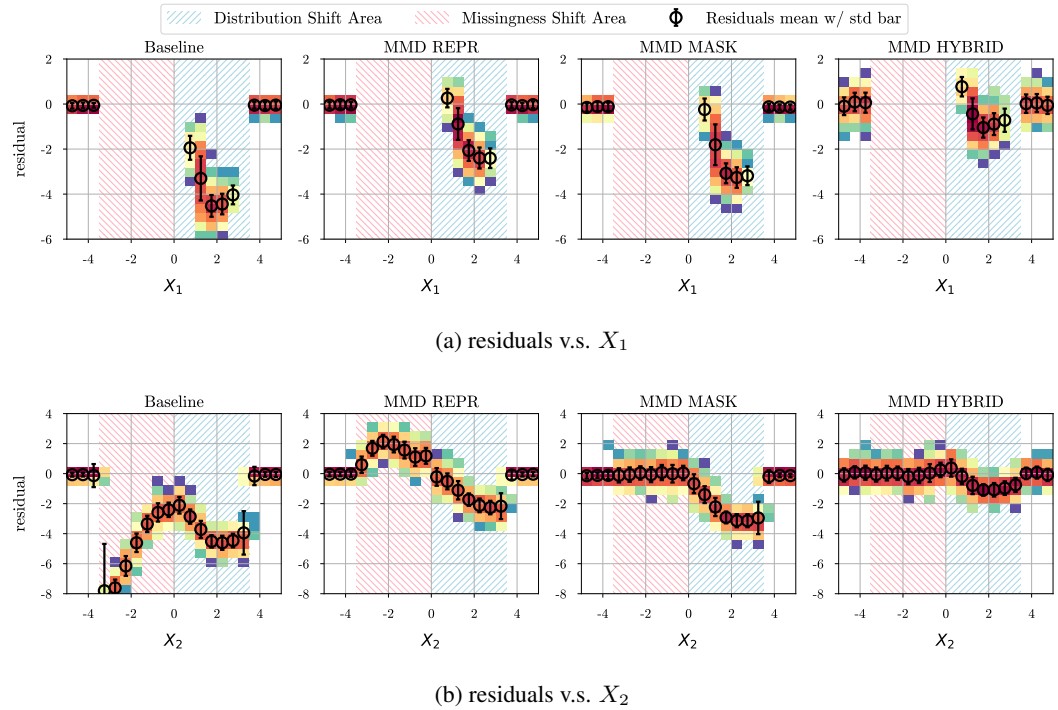

(a) residuals v.s. $X_1$

(b) residuals v.s. $X_2$

Figure 3: Synthetic data: residuals v.s. feature $X_1$ and $X_2$ on test set from the baseline model, MMD Representation model, MMD Mask model and MMD Hybrid model. The heatmap shows the sample frequency in each small cell. The black circles and bars represent the average and standard deviation of residuals in each bucket of $X_1/X_2$. Distribution shift of $X_1$ occurs in blue shaded area and missingness shift of $X_1$ occurs in pink shaded area.

Table 1: Averages and standard deviations of performance metrics (MSE for synthetic data, RMSE for the Bike Sharing and AUC for the IEEE-CIS Fraud Detection) of 10 runs on test data for each model.

| MODEL | SYNTHETIC DATA | BIKE SHARING | FRAUD DETECTION |
|-------|----------------|--------------|-----------------|
| BASELINE | $17.682 \pm 9.911$ | $116.2 \pm 3.9$ | $86.29\% \pm 0.59\%$ |
| KMM | $17.666 \pm 9.854$ | $116.5 \pm 5.2$ | N.A. |
| DAN | $0.753 \pm 0.698$ | $106.3 \pm 2.3$ | $85.91\% \pm 0.79\%$ |
| JAN | $0.820 \pm 0.693$ | $107.4 \pm 2.2$ | $86.30\% \pm 0.54\%$ |
| MMD REPR | $2.303 \pm 1.373$ | $107.7 \pm 1.7$ | $86.44\% \pm 0.48\%$ |
| MMD MASK | $2.573 \pm 1.119$ | $106.1 \pm 7.0$ | $87.25\% \pm 0.19\%$ |
| MMD HYBRID | $0.331 \pm 0.201$ | $98.7 \pm 3.8$ | $87.22\% \pm 0.32\%$ |
| GOLDEN MODEL | $0.180 \pm$ N.A. | N.A. | N.A. |

MMD Hybrid, in order. This indicates that MMD Mask can treat some distribution shift, but not as well as MMD Representation.

Furthermore, we checked the embeddings in the last hidden layer from the training and test phases for each model. Figure 4 shows 2-D tSNE (van der Maaten & Hinton, 2008) plots of embeddings from the baseline model, MMD Representation, MMD Mask and MMD Hybrid. As expected, in all MMD approaches, the learned embeddings between the training and test sets are more similar to each other than in the baseline model. Among all approaches, MMD Hybrid learned the most consistent embeddings between the training and test sets. Appendix B has more embedding results for KMM, DAN and JAN, none of which is better than MMD Hybrid.

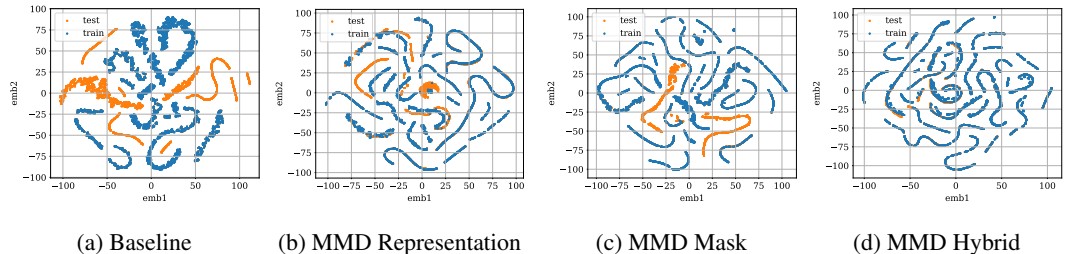

| (a) Baseline | (b) MMD Representation | (c) MMD Mask | (d) MMD Hybrid |

Figure 4: tSNE plot of embeddings from last hidden layer from each model on synthetic data.

Finally, we examined the masks generated by MMD Mask and MMD Hybrid. Figure 5a shows histograms of features $X_1$ in the original training set, the test set, and the masked training set. We focused on $X_1$ here because there is no shift in $X_2$. As we can see, the masker in MMD Mask and MMD Hybrid learned to mask out the shifted range on both the positive and negative sides to some degree. Since the negative side has purely a missingness shift, it learned better by masking out more samples for $X_1 \in [-3.5, 0]$, while it masked out less samples for $X_1 \in (0, 3.5]$. The masking ability is consistent with the downstream performance we observed.

## 4.2 BIKE SHARING AND IEEE-CIS FRAUD DETECTION

We use the Bike Sharing as the regression example and the IEEE-CIS Fraud Detection as the classification example to show how our MMD models perform in general. The Bike Sharing dataset (Fanaee-T & Gama, 2013) from UCI contains the hourly and daily bike rental counts in the Capital bikeshare system in Washington, D.C., USA for 2011 and 2012. We focused on hourly data as it provides more samples. The dataset contains features like hour, month, season, workday, holiday and some weather information. We log-transformed the target variable (bike rental counts) due to their long tail, and deliberately selected 2011-03 to 2011-11 as the training data (6,567 rows) and 2011-12 to 2012-03 as the test data (2,917 rows) so that the features have shifts between the training and test data due to the different times of year. Also, since there are some new months in the test set

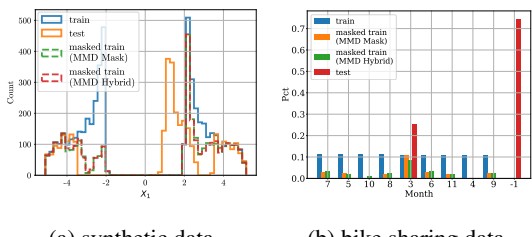

| (a) synthetic data | (b) bike sharing data |

Figure 5: Comparison of masked training data generated by MMD Mask and MMD Hybrid with original training and test data. (a) Synthetic data: histograms for feature $X_1$ in original train, test and masked train. (b) The Bike Sharing data: frequency percentage of each month in original train, test and masked training data. -1 represents the new months appearing in test set.

(December, January and February), after ordinal feature processing, they are treated as missing in the prediction stage, which can be viewed as a source of missingness shift.

The IEEE-CIS Fraud Detection dataset is from Vesta's real-world-e-commerce transactions and contains a wide range of features from device type to product features. The goal is to estimate fraud probability of the unlabeled test set. We followed the champion model from Chris Deotte and Konstantin Yakovlev[4], and used all the features except 219 Vesta engineered rich features that were determined redundant by their correlation analysis, but did not conduct further feature engineering like they did, as our goal is not to outperform the best model but use this dataset to show how our proposed MMD models can handle distribution and missingness shift in the features and outperform the baseline models. In the end, we used 212 features before one hot encoding the categorical features and adding missing indicators. Missingness shift is observed in some of the 212 features to some degree, but not severe.

Table 1 shows the averages and standard deviations of the performance metric (RMSE for the Bike Sharing and AUC for the IEEE-CIS Fraud Detection) on test data of 10 runs for each model. KMM performance is not available for IEEE-CIS Fraud Detection as it is infeasible to solve the quadratic problem using cvxopt package when feature dimension is high and sample size is large. For the Bike Sharing data, KMM does not improve the performance compared to the baseline. We think the

---

[4]https://www.kaggle.com/cdeotte/xgb-fraud-with-magic-0-9600.

fundamental reason is that importance re-weighting techniques such as KMM assume the support of test data is contained in the support of training data. However, this is not true for this test data. As we can see, all MMD approaches improved the model performance for both datasets. Because the Bike Sharing dataset contains both distribution shift and missingness shift, we see that the MMD Hybrid model works best, while DAN and JAN have similar performance compared to MMD Representation and MMD Mask. As for IEEE-CIS Fraud Detection data, we see MMD Representation style methods (MMD Representation, DAN, JAN) have comparable performance to the baseline. MMD Mask and MMD Hybrid perform better than others as they can handle missingness shift better.

We also examined if the masks learned for the month feature in the Bike Sharing dataset meet our expectation. As we can see from Figure 5b, the maskers learn to leave more of March 2011 data in while masking out data from other months more, which matches with our expectation, as March is the only common month in both training and test data. Note that the maskers try to match the joint distribution of all features, not just the marginal distribution of months.

The results on the two real world datasets are consistent with our findings in synthetic data. We also tested our methods on more unstructured data like MNIST to show they also work on complex image data. In the case where occlusion is correlated with features/labels on image data, MMD Mask and MMD Hybrid can work more efficiently compared to random data augmentation to learn the correct masks to use on the train images. Due to space limits, we leave the experiment details and results in Appendix C.

## 4.3 CARBON EMISSION ESTIMATION

The experiments in this section are designed to answer question (3). Carbon Emission Estimation is an internal project that seeks to estimate companies' carbon emissions using various sets of features including environmental, social and governance (ESG) data, industry segmentation data, fundamental financial data, and country and regional data. In total, there are about 1,000 features. One challenge of the project is the missingness shift between labeled and unlabeled data. The labeled data usually has more complete feature information, while unlabeled data has many missing values in different sets of features. As a result, a model trained on the labeled data would generate poor results on the unlabeled data due to the much larger missing rates in various features. We could not take the approach of dropping all features that are sometimes missing as nearly all features in this dataset can be missing. To overcome this challenge, we propose using data augmentation where we generate masks for the labeled data to mimic the missingness patterns in unlabeled data.

In this case, the downstream predictive model is a tree-based model without representation learning ability, so only MMD Mask can be applied here. We trained a masker model to generate masks for labeled data based on all available features by matching the MMD statistics between the labeled and unlabeled data. As a baseline, we also tried using a simple mask model that randomly sampled masks from unlabeled data conditioned on industry. The downstream carbon estimation model was then trained on the masked labeled data with 10 masks sampled from the mask model plus the original training data. We denote MMD Model as the downstream model trained on data masked by masks from the MMD Mask model, and Simple Model as the downstream model trained on data masked by masks from the simple mask model.

To evaluate the results, we designed a series of tests. First, we created a masked dataset by carrying over masks for companies from a year in which they did not report carbon emissions to a following year in which they did. This is the most realistic set of masks we can get for the labeled data and we considered it the golden set. Figure 6a shows the percentage error of prediction from the MMD Model and the Simple Model on this mask-carryover set. As we can see, the MMD Model has a large performance gain over the Simple Model.

We also tested the performance of the MMD Model and Simple Model on data masked by both MMD masks and simple masks. We sampled 10 masks from the corresponding mask model and applied those to the original labeled data, then compared predictions from the downstream model trained with two different versions of masked data. Figure 6b compares the percentage error of prediction for each combination. As we can see, the MMD Model always beats the Simple Model whether the data is masked by MMD mask or the simple mask. And when the original labeled data is masked by MMD mask, the performance gain is even larger. The fact that the MMD Model performs better than the Simple Model even on the data masked by the simple mask is interesting. It indicates that the

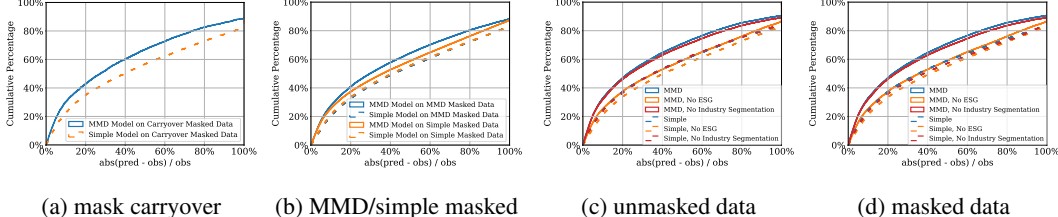

| (a) mask carryover | (b) MMD/simple masked | (c) unmasked data | (d) masked data |

Figure 6: arbon emission estimation: cumulative percentages for the percentage error of prediction. (a) MMD Model / Simple Model performance on mask-carryover set. (b) MMD Model / Simple Model performance on MMD Mask/simple mask masked data. (c) MMD Model / Simple Model performance on unmasked data when a set of feature is completely missing. (d) MMD Model / Simple Model performance on masked data when a set of feature is completely missing.

missingness pattern from the simple mask is in a sense contained in the missingness pattern from MMD mask.

In addition, we tested the MMD Model and Simple Model in the specific scenarios that an entire set of features is missing, as this can be the case for certain sub-universes of companies. Figure 6c and Figure 6d plots the scenarios on unmasked data and masked data respectively. Since ESG data is much more useful for estimating carbon emissions than industry segmentation data is, the performance decreases more when ESG data is missing. However, no matter which set of features is entirely missing, the MMD Model always performs better than the Simple Model. Also, if we compare the curves for unmasked data with those for masked data, we can see that most of them are very similar except for the Simple Model with Segmentation data missing. The Simple Model performance on masked data is a bit worse compared to unmasked data when Segmentation data is completely missing. However, MMD Model performance on masked data remains similar to that on unmasked data, indicating that MMD masks are more aligned with the scenarios when an entire set of features is missing.

## 5 CONCLUSION AND FUTURE WORK

We have demonstrated through various experiments that optimizing MMD statistics works well to build a more robust model when there are shifts between the training and test datasets. More importantly, we proposed a novel MMD masker model to match the joint distribution of input features and missingness indicators in input space between the training and test datasets, an approach designed specifically for treating a missingness shift, which has never been considered in previous UDA works. We have shown that MMD Representation works better for distribution shifts while MMD Mask works better for missingness shifts on both synthetic data and real data. Furthermore, we combined MMD matching in the feature embedding space and in the raw input space to yield superior results.

With the current approaches introduced in this paper, the learned model is calibrated to a specific test set. We think that MMD has the potential to learn a more generalized model that is robust for different sets of test data if we combine this technique with meta-learning approaches. This is out of scope for this paper and an interesting area for future work to explore. However, in scenarios where one has a target test set to label, our proposed MMD approaches help the model learn to use the best features in both the input and embedding spaces and avoid dangerous extrapolation as much as possible, and as a result, achieve better performance on the test set.

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

## A    MODEL ARCHITECTURES AND HYPERPARAMETERS DETAILS

For synthetic data, the baseline model is an MLP with 3 hidden layers with a hidden size of 16 for each. The MMD Mask model has 3 hidden layers with hidden sizes of $\{32, 32, 20\}$. For the MMD Representation and MMD Hybrid models, we also tested different hyperparameter values $\lambda \in \{1, 5, 10\}$ and picked the model that performed best. In all models, we used ReLU as our activation function and optimized network parameters using RMSProp with a learning rate of 0.01. All models were trained for 5,000 epochs with full batch.

For the Bike Sharing, the baseline model is an MLP model with 4 hidden layers with hidden size of $\{64, 64, 64, 64\}$, and the MMD Mask model has 3 hidden layers with hidden sizes of $\{512, 512, 128\}$. For the IEEE-CIS Fraud Detection, the baseline model is an MLP model with 4 hidden layers with hidden size of $\{1024, 512, 512, 256\}$, and the MMD Mask model has 3 hidden layers with hidden sizes of $\{512, 512, 256\}$. For both datasets, we used ReLU as our activation function and optimized network parameters using RMSProp. For the MMD Representation and MMD Hybrid models, we also tested different hyperparameter values $\lambda \in \{1, 10, 100\}$. The labeled data is split into 3:1 as train v.s. validation. The hyperparameters are tuned based on the performance on the validation set.

For carbon emission estimtion, the MMD Mask model is an MLP model with 4 hidden layers with hidden size of $\{512, 512, 512, 512\}$, with ReLU activation. We trained the model for 300 epochs with a batch size of 5000, using RMSProp with a learning rate of 0.001.

## B    MORE RESULTS ON SYNTHETIC DATA

In main sections, we took a scrutiny of MMD Representation, MMD Mask and MMD Hybrid in regions with different types of shift. Here we also include similar analysis for KMM, DAN and JAN. From Figure 7, we can see that KMM, DAN and JAN also have very small residuals in the region where data shift does not exist. When missingness shift or distribution shift exists, performance of all

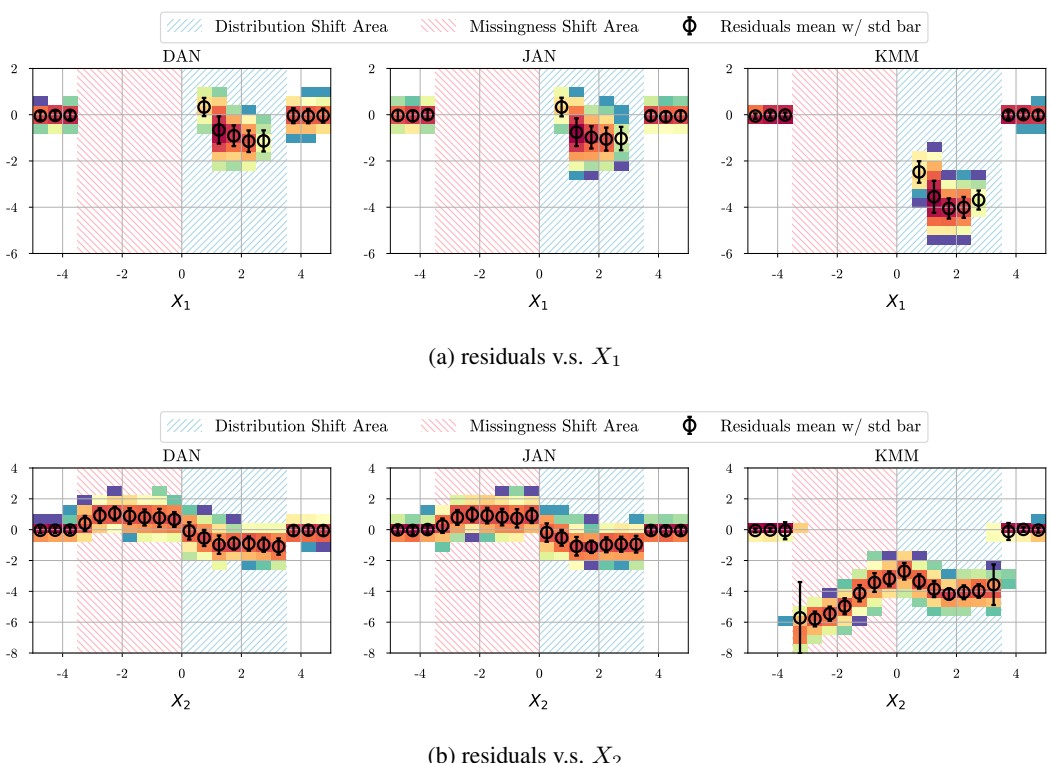

(a) residuals v.s. $X_1$

(b) residuals v.s. $X_2$

Figure 7: Synthetic data: residuals v.s. feature $X_1$ and $X_2$ on test set from the KMM, DAN and JAN. The heatmap shows the sample frequency in each small cell. The black circles and bars represent the average and standard deviation of residuals in each bucket of $X_1/X_2$. Distribution shift of $X_1$ occurs in blue shaded area and missingness shift of $X_1$ occurs in pink shaded area.

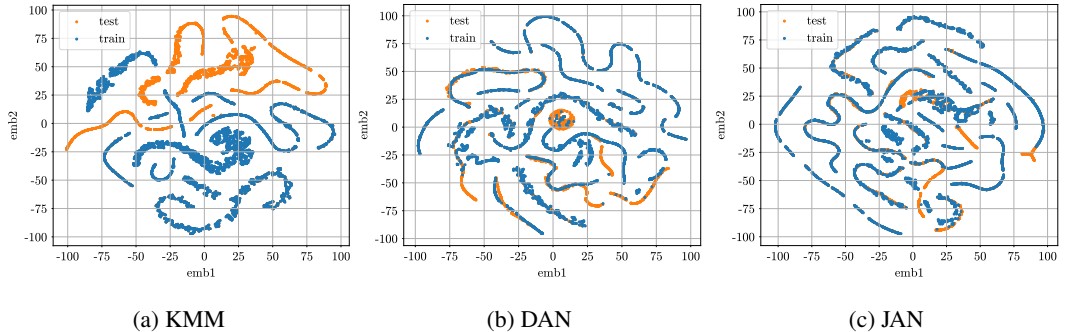

| (a) KMM | (b) DAN | (c) JAN |

Figure 8: tSNE plot of embeddings from last hidden layer from each model on synthetic data.

methods deteriorates to different degrees. KMM has a similar performance compared to the Baseline model in both missingness shift and distribution shift regions. In the region where distribution shift exists, DAN and JAN work better than MMD Mask, which is expected. DAN and JAN also work better than MMD Representation as they try to match more intermediate layers, and work similarly with MMD Hybrid. However, in the region where missingness shift exists, DAN and JAN are not as good as MMD Mask and MMD Hybrid, but similar to MMD Representation.

Further, we checked the embeddings in the last hidden layer from the training and test phases for KMM, DAN and JAN as well. Figure 8 shows 2-D tSNE plots of embeddings from each model. There is very clear separation between training and test embeddings from KMM, similarly to the baseline model. DAN and JAN both make the training and test embeddings more similar to each other, but still not as similar as those generated by our MMD Hybrid.

## C  MNIST EXPERIMENTS

In this section, we investigate how our MMD methods perform on image data. Our proposed methods can be useful when occlusion is correlated with features/labels on image data. For example, say you have intact images of groundhogs, squirrels and pangolins with labels in training set. But in the unlabeled set, the images of groundhogs and pangolins are incomplete as those are taken when the animals are partially in ground. So we need to learn appropriated masks to mask out right images in training set in order to learn a more accurate and robust classification model.

Table 2: The model architecture for the baseline model in MNIST experiments. It outputs the logits of digits class.

| |
| --- |
| IMAGE $x \in \mathcal{R}^{M \times M \times 1}$ |
| $3 \times 3, stride = 1$ CONV 16 ReLU |
| $2 \times 2$, MAXPOOL |
| $3 \times 3, stride = 1$ CONV 32 ReLU |
| $2 \times 2$, MAXPOOL |
| DENSE $\rightarrow 10$ |

To demonstrate our approaches also work on high dimensional image data, we used MNIST which is a handwritten digits dataset where each image is $28 \times 28$ pixels. It has 60,000 training images and 10,000 test images. To demonstrate that our methods work when there is a shift, particularly a missingness shift, we created a patterned mask on the original test data. That is, for digits in $\{1, 4, 5, 7\}$, we mask out the last 12 rows of pixels, and for digits in $\{0, 2, 3, 6, 8, 9\}$, we mask out the last 12 columns of pixels.

We compared MMD Representation, MMD Mask, and MMD Hybrid approaches with a baseline model trained on raw training data, a golden model trained with the training samples masked by true masks, as well as traditional data augmentation. All models in this section adopt CNN architectures. The baseline model in MNIST experiments uses the architecture shown in Table 2. The golden model and two variants of data augmentation models also use the exactly same architecture. The MMD Representation uses the same architecture except that it also matches the embeddings between

Table 3: The model architecture for the MMD Mask in MNIST experiments. It outputs the logits of mask for each pixel.

| |
|---|
| IMAGE $x \in \mathcal{R}^{M \times M \times 1}$ |
| $3 \times 3, stride = 1$ CONV 16 ReLU |
| $3 \times 3, stride = 2$ CONV 16 ReLU |
| $3 \times 3, stride = 1$ CONV 32 ReLU |
| $3 \times 3, stride = 2$ CONV 32 ReLU |
| DENSE |
| $4 \times 4, stride = 2$ DECONV 32 ReLU |
| $3 \times 3, stride = 1$ DECONV 32 ReLU |
| $4 \times 4, stride = 2$ DECONV 16 ReLU |
| $3 \times 3, stride = 1$ DECONV 16 LINEAR $\rightarrow$ 784 |

Table 4: Mean and standard deviation of test accuracy from 10 runs for MNIST.

| MODEL | ACC MEAN ± STD |
|---|---|
| BASELINE | $0.873 \pm 0.030$ |
| RANDOM ERASING - V1 | $0.934 \pm 0.008$ |
| RANDOM ERASING - V2 | $0.950 \pm 0.004$ |
| MMD REPR | $0.959 \pm 0.005$ |
| MMD MASK | $0.972 \pm 0.002$ |
| MMD HYBRID | $0.972 \pm 0.001$ |
| GOLDEN MODEL | $0.987 \pm 0.001$ |

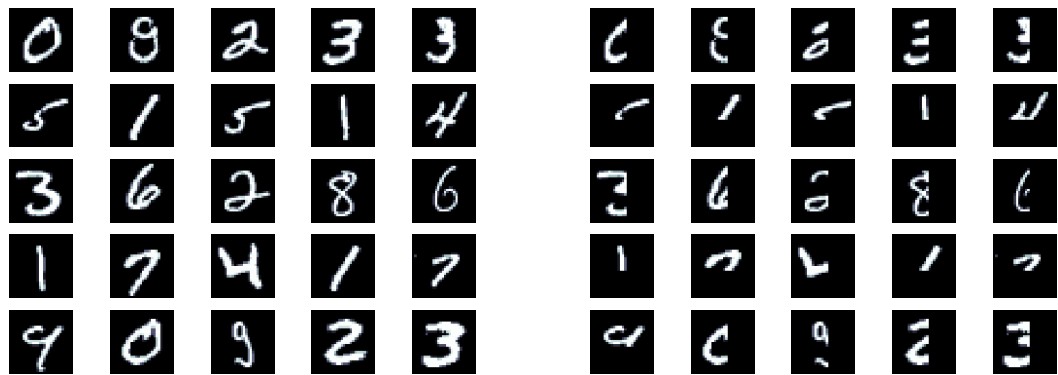

(a) Original training samples        (b) Masked training samples

Figure 9: Original training samples v.s. the training samples masked by the MMD Hybrid model.

training and test set before entering the final dense layer. The MMD Mask model generates masks using a conv-deconv architecture shown in Table 3. It generates masks to mask original training data, then the masked training data goes through the same downstream model. The MMD Hybrid model combines MMD Representation and MMD Mask. The missing values were imputed as zero (black pixel).

For traditional data augmentation, we tested two versions of RandomErasing (Zhong et al., 2020), with one version always erasing the same area ($12 \times 28$) of images at a random place and the other erasing a random area ranging from 0 to the same area ($12 \times 28$) at a random place. We followed the standard procedure to split the training samples into a training set of 54,000 rows and a validation set of 6,000 rows, and trained all models for 20 epochs with a batch size of 64 (except for the MMD Mask model, which was trained for 600 epochs with a larger batch size of 10,000). The hyperparameter $\lambda$

in the MMD Representation and MMD Hybrid model was set to 1. The final model in all methods was the one that gave the best validation loss.

The results are summarized in Table 4. The performance reported is the average test accuracy from 10 runs for each model. Note that this baseline model can achieve around 98% test accuracy on the original test dataset, but only around 87% accuracy on the corrupted test dataset. However, MMD Representation, MMD Mask, and MMD Hybrid can all improve the performance by a large margin, to 95.9%, 97.2%, and 97.2% accuracy, respectively. Because we corrupted the test set by applying patterned masks, MMD Mask is a more natural treatment compared to MMD Representation. Since MMD Hybrid combines MMD Representation and MMD Masks, it works as well as MMD Mask in this case. All three methods performed better than the two traditional data augmentation methods: these achieve accuracies of 93.4%and 95.0%. We believe that this is simply because the learned transformation is more effective and efficient compared to more random transformations. The golden model achieves 98.7% accuracy, which is not too far away from performances from the best MMD approaches. Figure 9 shows some samples of generated masks applied to the original training set, which are quite close to the true masks we imposed on the test set.

