# OpenReview forum: "Maximum Mean Discrepancy for Generalization in the Presence of Distribution and Missingness Shift"
_ICLR.cc/2022/Conference — ICLR 2022 Submitted_

### Official Review · Reviewer_jTXA · 2021-10-21

**Correctness:** 3
**Technical Novelty And Significance:** 2
**Empirical Novelty And Significance:** 2
**Recommendation:** 3
**Confidence:** 3

**Main Review:**

The paper addresses an important problem, proposes an interesting approach and I enjoyed reading this work. Furthermore, it pushes the idea of thinking about the distributions of feature representations under covariate-shift and of an end-to-end learning workflow (vs. common two-step approaches in the liteature). However, I also have some concerns, which I outline below.


# Related work on missing data:
At least half of the contribution of the paper pertains to supervised learning with missing data, yet no connection to existing approaches is made. Some example recent references include the following (also cf. the work of Prof. Julie Josse and statistical literature on missingness more broadly)

> Smieja, M., Struski, Ł., Tabor, J., Zieliński, B., & Spurek, P. (2018, December). Processing of missing data by neural networks. In Proceedings of the 32nd International Conference on Neural Information Processing Systems (pp. 2724-2734).

> Ipsen, N., Mattei, P. A., & Frellsen, J. (2020, July). How to deal with missing data in supervised deep learning?. In ICML Workshop on the Art of Learning with Missing Values (Artemiss).

> You, J., Ma, X., Ding, D. Y., Kochenderfer, M., & Leskovec, J. (2020). Handling missing data with graph representation learning. arXiv preprint arXiv:2010.16418.

I think it is important to put this work into this broader context of research on missing data problems. Furthermore, in any discussion of missing data it is important to outline what missing data one anticipates (or what type of missing data the method would perform well for), is it data missing at random, is the missingness informative, etc.

# Evidence for the performance of proposed approaches:

After reading the paper, I do not know when I would try to use this approach in practice. The reason is that this work does not explain either (let's say by formal theoretical results, or heuristics), when the method is expected to perform well, and when existing approaches would be preferable (also see the point above about assumptions on missingness). Also since the only evidence provided is in terms of empirical results, I think these would need to be more comprehensive to provide insight to a reader (e.g. more ablation studies, what is the impact of learning model architectures, more settings, etc).


# Details of method and reproducibility:

While reading the paper, I often felt lost. I do not think I would be able to reproduce the results based on just the description in the text. I think this could be alleviated by providing more explicit detail. Some concrete questions I had:

* I would have liked some more detail about the methods used to generate the masking scheme. What happens if the mask specifies that an entry that is currently missing in the training set should be unmasked? I thought for some time that maybe the assumption is that the training set has no missing data (only the test set has), but this seems not to be the case in Section 4.3. Is the assumption that there is a stochastic dominance with strictly more missingness in the test data? How do the models predict with the masked data? Is missing data treated as a "zero" by e.g. the linear combinations of the neural net layers?

* "Generally we found that setting $\lambda$ to match the magnitudes of the two loss terms leads to the best results": Could you elaborate on this step?

* Algorithm 1 box: Are the update steps minimization steps? Or gradient updates? Is there a typo in the update step for $f_M$, i.e., should there be $(\mathbf{X}'\_{tr}, \hat{\mathbf{I}})$ instead of  $(\mathbf{X}\_{tr}, \hat{\mathbf{I}})$?

* Simulation: Could you provide more detail on the data generation? Is the model an additive model, namely the sum of the univariate responses shown in terms of $X_1$ and $X_2$? How are $X_1$ and $X_2$ generated? I thought they would be independent, but this contradicts Figure 3, where the range in which $X_1$ is missing also corresponds to a deterioration in residuals in terms of $X_2$.



**Summary Of The Paper:**

Suppose we are given training data  $(\mathbf{X}_{tr}, \mathbf{I}\_{tr},  \mathbf{Y}\_{tr})$, where $\mathbf{Y}\_{tr} \in \mathbb R^{n\_{tr}}$  is the response of interest, $\mathbf{X}\_{tr} \mathbb R^{n\_{tr} \times p}$ is the feature design matrix ($p$ features, $n\_{tr}$ samples) and  $\mathbf{I}\_{tr} \in  \\{0,1\\}^{n\_{tr} \times p}$ is an indicator matrix that marks which features are missing in $\mathbf{X}\_{tr}$. In the setting studied in this paper, the researcher also has access to the test samples  $(\mathbf{X}\_{te}, \mathbf{I}\_{te})$ where $\mathbf{X}\_{te} \in \mathbb R^{n\_{te} \times p}$ and $\mathbf{I}\_{te} \in  \\{0,1\\}^{n\_{te} \times p}$ are the features of the test samples and their missingness indicator. The goal is to predict the response of the test samples while accounting for the potential complications of covariate-shift and a shift in the missingness pattern between training and test set covariates.

The authors propose three approaches to this general problem.

1) MMD-Representation to account for covariate-shift (ignoring missingness): here one learns a feature representation $f_F$ such that the distribution of $f_F(x\_{tr})$ and  $f_F(x\_{te})$ are similar, where $x\_{tr}$ is a feature vector drawn from the training distribution and $x\_{te}$ from the test distribution. Closeness here is measured in terms of the maximum mean discrepancy of Gretton and colleagues (2007). Then one learns a predictive model $f_P$ so that the predictions take the form $f_P(f_F(x))$.  The two models $f_P$ and $f_F$ are learned simultaneously by minimizing a weighted combination of the predictive training loss and the MMD loss between test and train featurization.

2) MMD-mask to deal with varying missingness patterns: here on learns a masking model $f_M$ and applies it as $\hat{\mathbf{I}} = f_M(\mathbf{X}_{tr}, \mathbf{I}\_{tr})$ to learn a missingness pattern that matches more closely the missingness pattern of the test dataset. Then one applies the learned missingness pattern on the training samples to get modified training samples $\mathbf{X}'\_{tr}$ with missingness pattern $\hat{\mathbf{I}}$. The masking model is learned by minimizing the maximum mean discrepancy between the distributions of  $(x'\_{tr}, \hat{i}\_{tr})$ and $(x\_{te}, i\_{te})$. Finally one learns the predictive model based on  $(\mathbf{X}'\_{tr}, \hat{\mathbf{I}}, \mathbf{Y}\_{tr})$.

3) MMD-hybrid: a combined approach of MMD-mask and MMD-representation to account for both shifts in the missingness pattern and covariate-shift.

These three approaches are benchmarked and compared in one synthetic dataset and two applications (Bike Sharing and IEEE-CIS Fraud Detection).






**Summary Of The Review:**

The paper addresses an important topic and is interesting. Nevertheless, at this time I vote to reject it for the following reasons:

* There is not enough intuition/theory regarding situations in which the method is expected to perform well compared to existing work, nor are the empirical results sufficiently detailed.

* Related work on supervised learning with missing data is completely omitted.

* The authors would need to provide additional detail to facilitate the reproducibility of the results.

---

### Official Review · Reviewer_UfTS · 2021-11-02

**Correctness:** 4
**Technical Novelty And Significance:** 2
**Empirical Novelty And Significance:** 2
**Recommendation:** 3
**Confidence:** 5

**Main Review:**

The manuscript is easy to follow. The proposed methodology to address both missingness and distributional shift makes sense. Effectiveness of the proposed methodology has also been validated in real-world datasets.

However, I am still worried about the novelty of the proposed methodology. In fact, use MMD to match latent distributions is not novel at all. As authors have mentioned, similar or almost the same strategy has been used in previous literature like DAN (Long et al., 2015) and JAN (Long et al., 2017).

Learnng a new mask on training data to mimic the missingness patterns in test data seems an interesting idea. However, I still feel this contribution is marginal. In practice, I only have one question: how to properly set the hyperparameters (like $\lambda$) of both MMD mask and MMD representation in the joint loss function?


**Summary Of The Paper:**

Authors suggest a new MMD based methodology to improve generalization in case of both distributional shift and missing data. For the missing data, authors propose to generate a new mask (or missingness indicator) on the training data to mimic the missingness patterns in the test data. The closeness of missing patterns in training and test data can be measured with MMD.

**Summary Of The Review:**

My major concern is the novelty of this work. Aligning distributions of latent representations by MMD has been extensively investigated before. It seems to me the only technical contribution is to generate a mask that mimics the missing patterns in test data.

---

### Official Review · Reviewer_x6L8 · 2021-11-02

**Correctness:** 2
**Technical Novelty And Significance:** 1
**Empirical Novelty And Significance:** 2
**Recommendation:** 3
**Confidence:** 4

**Main Review:**

*Strengths*
- The paper considers model adaptation under different missing patterns in the source and target domains and/or covariate shift.
- Proposed algorithm based on MMD and intensive empirical evaluations

*Weaknesses*
- The paper lacks a formal definition of the problem and proposed solution. For instance, several works were carried out on missing data imputation under various settings (missing at random, missing completely at random...) and are not discussed in the paper. The MMD mask model is not formaly defined.

- The proposed method lacks theoretical analysis on the test error bound when adapting the model.

- The paper overlooks a closedly related work [1] which is more elaborated and more principled. In my opinion, compared to [1] the proposed contribution is less significant.


*References*

[1] Kirchmeyer, Matthieu, et al. "Unsupervised domain adaptation with non-stochastic missing data." Data Mining and Knowledge Discovery (2021): 1-42.

*After rebuttal*

No feeback was provided by the authors. Hence, the review and the paper evaluation remain the same.

**Summary Of The Paper:**

This work proposes a solution dealing with model adaptation under covariate shift and/or missingness shift  by minimizing the Maximum Mean Discrepancy (MMD) between the training labeled set and unlabeled test set. The method is supported by several evaluations on synthetic data and real data.

**Summary Of The Review:**

The paper proposes a methodological framework to study DA under missingness and covariate shifts. It lacks a clear formulation, a theoretical analysis and it overlooks existing methods. Hence, the contribution is not significative.

---

### Official Review · Reviewer_4NFR · 2021-11-05

**Correctness:** 3
**Technical Novelty And Significance:** 2
**Empirical Novelty And Significance:** 2
**Recommendation:** 3
**Confidence:** 2

**Main Review:**

Pros:
- Considering both distribution shifts and missingness is an important direction of research that can improve practical performances of modern machine learning models.
- The paper is easy to follow and well-structured.
- The method provides improved results on small benchmark datasets.

Cons:
- Authors limit their scope in the situation that they are given all test inputs at the training phase. But this seems to be too restricted scope, considering the majority of the community works on robust machine learning models that can cope with novel distribution shifts patterns.
- Regarding this point, I also have a concern about potential data snooping problems. Specifically, using the test inputs to regularize the feature during training seems to be analogous to using them in the semi-supervised learning loss functions. For this reason, I suggest authors compare their methods with several semi-supervised learning methods such as MixMatch and VAT. In current form, all experiments put too much favor to MMD-based models compared to other baseline.
- The authors did not use an approximate MMD distance, so the proposed method has limited scalability. Also, due to the implementation of MMD in the paper, the authors compare all methods using full-batch training, which is highly unlikely in practice and standard benchmark problems. For this reason, I’d recommend authors to add the baseline trained with standard training schemes involving mini-batch SGD.



**Summary Of The Paper:**

This paper considers distribution shift and feature missingness problems. To solve this, the authors minimize MMD distance between training and test samples. Also, to make representation robust to novel missing patterns in test sets, the authors propose to learn making operations so that the resulting representation has small MMD distance. Authors verify their methods in synthetic dataset and small benchmark problems.


**Summary Of The Review:**

The paper has a limited applicability to large-scale problems. Considering the modern methods and tricks has already dealt with the large-scale problems on a more diverse set of distribution shifts (such as Hendrycks 2019 ICLR), the results in the paper seem not significant. Also, the experiments can be performed more precisely if they involve more proper baselines.

---

### Decision · Program_Chairs · 2022-01-20

**Decision:**

Reject

**Comment:**

The paper addresses unsupervised domain adaptation under covariate shift and missing source and target features. Three approaches are proposed for tackling respectively covariate shift, missing data and simultaneous covariate shift and missing data. The proposed method relies on the minimization of the maximum mean discrepancy between the source and target representations in the different settings. Experiments are performed on a synthetic dataset and on two other datasets.

All the reviewers highlighted several weaknesses: lack of formal definitions and of formal analyses, lack of connection with existing approaches for handling missing data, weak reproducibility. The authors did not provide responses. Reject.